# Effect of SGLT-2 inhibitors on body composition in patients with type 2 diabetes mellitus: A meta-analysis of randomized controlled trials

**Runzhou Pan**, **Yan Zhang, Rongrong Wang, Yao Xu, Hong Ji, Yongcai Zhao** *

Department of Endocrinology, Cangzhou Central Hospital, Cangzhou, Hebei Province, China

* yongcaizhao2022@163.com

**Data Availability Statement:** All relevant data are within the paper and its Supporting Information files.

## Abstract

### Objective

Type 2 diabetes mellitus(T2DM) is closely related to sarcopenic obesity(SO). Body composition measurement including body weight, body mass index, waist circumference, percentage body fat, fat mass, muscle mass, visceral adipose tissue and subcutaneus adipose tissue, plays a key role in evaluating T2DM and SO. The weight reduction effect of sodium-glucose cotransporter 2(SGLT-2) inhibitors has been demonstrated. However, there are warnings that SGLT-2 inhibitors should be used with caution because they may increase the risk of sarcopenia. The effect of SGLT-2 inhibitors on body composition in T2DM is inconclusive. In this work, a meta-analysis of randomized controlled trials was conducted to evaluate the effect of SGLT-2 inhibitors on body composition in T2DM.

### Methods

PubMed, the Cochrane Library, EMbase and Web of Science databases were searched by computer. All statistical analyses were carried out with Review Manager version 5. 3. Results were compared by weight mean difference(WMD), with 95% confidence intervals (CI) for continuous outcomes. A random effects model was applied regardless of heterogeneity. The $I^2$ statistic was applied to evaluate the heterogeneity of studies. Publication bias was assessed using Funnel plots.

### Results

18 studies with 1430 participants were eligible for the meta-analysis. SGLT-2 inhibitors significantly reduced body weight(WMD:-2. 73kg, 95%CI: -3. 32 to -2. 13, p<0. 00001), body mass index(WMD:-1. 13kg/m$^2$, 95%CI: -1. 77 to -0. 50, p = 0. 0005), waist circumference (WMD:-2. 20cm, 95%CI: -3. 81 to -0. 58, p = 0. 008), visceral fat area(MD:-14. 79cm$^2$, 95% CI: -24. 65 to -4. 93, p = 0. 003), subcutaneous fat area(WMD:-23. 27cm$^2$, 95% CI:-46. 44 to -0. 11, P = 0. 05), fat mass(WMD:-1. 16kg, 95%CI: -2. 01 to -0. 31, p = 0. 008), percentage body fat(WMD:-1. 50%, 95%CI:-2. 12 to -0. 87, P<0. 00001), lean mass(WMD:-0. 76kg,

**Funding:** The author(s) received no specific funding for this work.

**Competing interests:** The authors have declared that no competing interests exist.

95%CI:-1. 53 to 0. 01, P = 0. 05) and skeletal muscle mass(WMD:-1. 01kg, 95%CI:-1. 91 to -0. 11, P = 0. 03).

## Conclusion

SGLT-2 inhibitors improve body composition in T2DM including body weight, body mass index, waist circumference, visceral fat area, subcutaneous fat area, percentage body fat and fat mass reduction, but cause adverse effects of reducing muscle mass. Therefore, until more evidence is obtained to support that SGLT-2 inhibitors increase the risk of sarcopenia, not only the benefit on body composition, but also the adverse effect of the reduction in muscle mass by SGLT-2 inhibitors in T2DM should be considered.

## Introduction

As one of the most serious and pressing health problems worldwide, type 2 diabetes mellitus (T2DM) is closely correlated with obesity, usually assessed by body mass index(BMI). However, BMI has crucial limitations due to its inability to assess weight distribution, fat mass and muscle mass. Fat accumulation and lean mass decrease are important changes that occur as adults age and are associated with an increased risk of T2DM [1]. T2DM is more closely related to fat distribution, percentage body fat and skeletal muscle than BMI [2]. Sarcopenic obesity (SO) is a new kind of complex syndrome characterized by double burden of sarcopenia (low muscle mass, muscle strength decreases and physical dysfunction) and excess fat [3]. T2DM is closely related to SO, and obesity is a common risk factor for both [4]. Body composition measurements include body weight, BMI, waist circumference, fat mass, percentage body fat, muscle mass, visceral adipose tissue and subcutaneus adipose tissue, which is performed by non-invasive techniques, playing a key role in evaluating T2DM and SO [5, 6].

As new class of oral hypoglycemic agents, sodium-glucose cotransporter 2(SGLT-2) inhibitors are paid attention due to unique mechanism of inhibiting proximal tubular glucose reabsorption and increasing urinary glucose excretion, which have been demonstrated to reduce body weight, improve cardiovascular and renal outcomes [7, 8]. However, there are warnings that SGLT-2 inhibitors should be used with caution due to the potential to increase the risk of dehydration and sarcopenia [9]. Therefore, the effect of SGLT-2 inhibitors on body composition in T2DM is worth discussing. This comprehensive systematic review and meta-analysis of randomized controlled trials (RCTs) aimed to evaluate the effects of SGLT-2 inhibitors on body composition in T2DM.

## Materials and methods

### Search strategy and study selection

PubMed, The Cochrane Library, EMbase and Web of Science databases were searched by computer. The combination of subject words and free words was used in the search. The search terms were as follows: (Sodium-Glucose Transporter 2 Inhibitors OR Ertugliflozin OR Dapagliflozin OR Canagliflozin OR Empagliflozin OR Ipragliflozin OR Luseogliflozin OR Tofogliflozin OR Sotagliflozin OR Gliflozins) AND (Body composition OR Waist circumference OR Skeletal muscle mass OR Fat mass OR Lean mass OR Visceral adipose tissue OR Subcutaneus adipose tissue), eligible search was limited to randomized controlled trials (RCTs). Two reviewers independently selected relevant articles based on their titles and abstracts, then

screened the full text and resolved any differences by consensus with a third reviewer to determine whether it met the inclusion or exclusion criteria. All RCTs evaluating SGLT-2 inhibitors on body composition in T2DM were included in the meta-analysis. The following inclusive selection criteria were applied: (1)Participants were clinically diagnosed with T2DM. Patients with type 1 diabetes and gestational diabetes were excluded. There were no restrictions on the age, sex or race of participants. (2)The treatment group consisted of various types of SGLT-2 inhibitors, meanwhile the control group consisted of other hypoglycemic drugs. Both groups had sufficient baseline and post-treatment information in the study report, such as body weight(BW), BMI, waist circumference(WC), percentage body fat(PBF), fat mass(FM), lean mass(LM), skeletal muscle mass(SMM), visceral fat area(VFA) and subcutaneus fat area(SFA). (3)The study design was RCTs. Case reports, animal experiments, conference abstracts, reviews, subgroup analysis and editorials were excluded.

## Data extraction and quality assessment

Two independent reviewers extracted the following information from eligible articles: first author, year of publication, sample size, type and dose of SGLT-2 inhibitors, control group medication, follow-up time and baseline patient information. Data collection for the following clinical outcomes: BW, BMI, FM, WC, PBF, LM, SMM, VFA and SFA. Two reviewers independently assessed the quality of RCTs using the Cochrane Risk of Bias tool, which includes the following seven criteria: random sequence generation (selection bias), allocation concealment(selection bias), blinding of participants and personnel(performance bias), blinding of outcome data(detection bias), incomplete outcome data(attrition bias), selective reporting (reporting bias)and other bias(certain biases not indicated above but influence the results). Each item was assessed as a "low risk", "high risk" or "unclear risk" of bias, according to recommendations in the Cochrane manual.

## Statistical analysis

All statistical analyses were carried out with Review Manager version 5.3 Results compared by Weight mean difference (WMD), with 95% confidence intervals(CI) for continuous outcomes. The $I^2$ statistic was used to assess study heterogeneity. Studies with $I^2$ statistic of 25%-50% were characterized as low heterogeneity, $I^2$ statistic of 50%-75% was characterized as moderate heterogeneity, $I^2$ statistic higher than 75% was characterized as high heterogeneity. A random effects model was applied regardless of heterogeneity, followed by subgroup analysis or sensitivity analysis to explain the reason for heterogeneity as soon as possible. Publication bias was assessed using Funnel plots.

## Results

### Literature search

A total of 473 articles were selected by preliminary search, with 73 articles being duplications. 400 records were eliminated based on the titles and abstracts. 171 full text of potential studies were retrieved for further evaluations. 153 articles were excluded, including 109 non-RCT studies, 31 conference abstracts, 13 incomplete trial studies. Finally, 18 studies were eligible for the meta-analysis. The selection process is shown in Fig 1.

### Basic characteristics and quality assessment

The characteristics of the 18 included studies published between 2013 and 2022 are shown in Table 1 [10–27]. The articles involved 1430 participants (726 SGLT-2 inhibitors participants

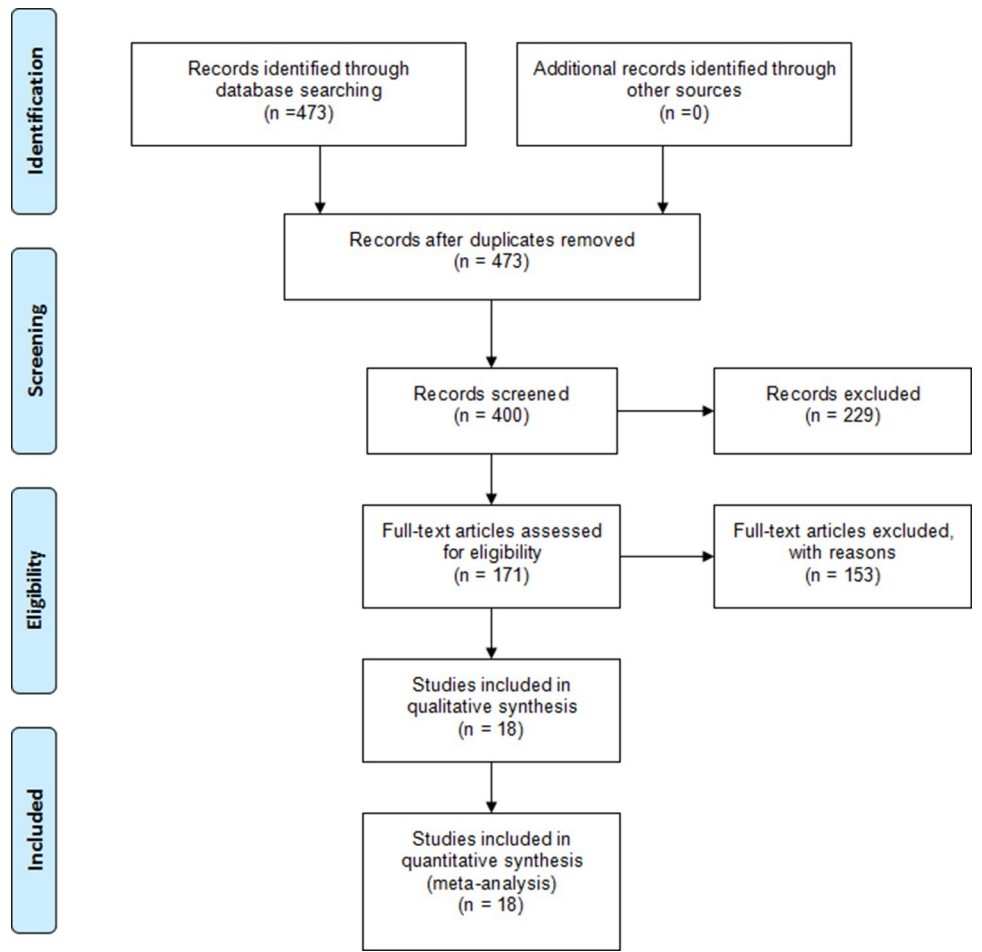

**Fig 1. Flowchart of studies included in this meta-analysis.**

and 704 control participants), including 7 multi-centre studies and 11 single-center studies. SGLT-2 inhibitors groups include canagliflozin (3 studies), dapagliflozin (8 studies), empagliflozin (3 studies) and ipragliflozin(4 studies), meanwhile control groups include traditional hypoglycemic drugs(16 studies), such as sulfonylureas, insulin, metformin, thiazolidinediones, DPP-4 inhibitors, etc and another new class of hypoglycemic drugs, GLP-1 receptor agonists (GLP-1RAs)(2 studies). 3 studies lasted 12 weeks, 10 studies lasted 24 weeks, 1 study lasted 26 weeks, 1 study lasted 28 weeks, 2 studies lasted 52 weeks and 1 study lasted 102 weeks. In addition, the Cochrane Risk Bias Assessment Tool was used to assess study bias. All studies were high-quality parallel grouped studies, and the quality assessment results of the included studies are summarized in S1 and S2 Figs.

## Outcome meta-analysis

**BW, BMI and WC.** Fourteen RCTs reported BW in 617 SGLT-2 inhibitors users and 604 non-users. The meta-analysis showed that SGLT-2 inhibitors treatment significantly decreased BW compared with other drugs (WMD:-2. 73kg, 95%CI: -3. 32 to -2. 13, p<0. 00001). Low heterogeneity was found between studies ($I^2$ = 33%). Seven RCTs reported BMI in 234 SGLT-2 inhibitors users and 219 non-users. In this meta-analysis, SGLT-2 inhibitors treatment significantly decreased BMI compared with other drugs (WMD:-1. 13kg/m$^2$, 95%CI: -1. 77 to -0.

**Table 1. Demographic and clinical characteristics of included studies.**

| Author | Country | Year published | Trial registration | Agent | Comparator | SGLT-2 inhibitors(n) | Control (n) | Follow-up time |
|---|---|---|---|---|---|---|---|---|
| Bode B et al [10] | 17 countries | 2013 | NCT01106651 | Traditional hypoglycemic treatment+Canagliflozin | Traditional hypoglycemic treatment +Placebo | 71 | 74 | 26 weeks |
| Cefalu WT et al [11] | 19 countries | 2013 | NCT00968812 | Canagliflozin+Metformin | Glimepiride+Metformin | 102 | 96 | 52 weeks |
| Bolinder J et al [12] | 5 countries | 2014 | NCT00855166 | Dapagliflozin+Metformin | Placebo+Metformin | 69 | 71 | 102 weeks |
| Fadini GP et al [13] | Italy | 2017 | NCT02327039 | Traditional hypoglycemic treatment+Dapagliflozin | Traditional hypoglycemic treatment +Placebo | 15 | 16 | 12 weeks |
| Ito D et al [14] | Japan | 2017 | UMIN000022651 | Traditional hypoglycemic treatment+Iapagliflozin | Traditional hypoglycemic treatment +Pioglitazone | 32 | 34 | 24 weeks |
| Han E et al [15] | Korea | 2020 | NCT02875821 | Ipragliflozin+Metformin +Pioglitazone | Metformin+Pioglitazone | 29 | 15 | 24 weeks |
| McCrimmon RJ et al [16] | 11 countries | 2020 | NCT03136484 | Canagliflozin+Metformin | Semaglutide+Metformin | 23 | 22 | 52 weeks |
| Nakaguchi H et al [17] | Japan | 2020 | UMIN000027614 | Traditional hypoglycemic treatment+Empagliflozin | Traditional hypoglycemic treatment +Liraglutide | 31 | 30 | 24 weeks |
| Yamakage H et al [18] | Japan | 2020 | UMIN000021479 | Traditional hypoglycemic treatment+Dapagliflozin | Traditional hypoglycemic treatment | 26 | 24 | 24 weeks |
| Wolf VLW et al [19] | Brazil | 2021 | NCT02919345 | Dapagliflozin+Metformin | Glibenclamide+ metformin | 44 | 45 | 12 weeks |
| Chehrehgosha H et al [20] | Iran | 2021 | IRCT20190122042450N3 | Empagliflozin | Pioglitazone | 35 | 34 | 24 weeks |
| Horibe K et al [21] | Japan | 2022 | UMIN000020239 | Traditional hypoglycemic treatment+Dapagliflozin | Traditional hypoglycemic treatment | 26 | 24 | 24 weeks |
| Inoue H et al [22] | Japan | 2019 | UMIN000018839 | Traditional hypoglycemic treatment+Ipragliflozin | Traditional hypoglycemic treatment | 24 | 24 | 24 weeks |
| Kayano H et al [23] | Japan | 2020 | UMIN000023834 | Traditional hypoglycemic treatment+Dapagliflozin | Traditional hypoglycemic treatment | 36 | 38 | 24 weeks |
| Kinoshita T et al [24] | Japan | 2020 | UMIN000021291 | Dapagliflozin | Pioglitazone | 32 | 33 | 28weeks |
| Tsurutani Y et al [25] | Japan | 2018 | UMIN000014738 | Traditional hypoglycemic treatment+Ipragliflozin | Traditional hypoglycemic treatment | 52 | 49 | 12 weeks |
| Shimizu M et al [26] | Japan | 2019 | UMIN000022155 | Traditional hypoglycemic treatment+Dapagliflozin | Traditional hypoglycemic treatment | 33 | 24 | 24 weeks |
| Zeng Y et al [27] | Taiwan | 2022 | NCT03458715 | Empagliflflozin +Premixed insulin | Linagliptin+Premixed insulin | 46 | 51 | 24 weeks |

50, p = 0. 0005). No heterogeneity was observed between studies ($I^2$ = 0%). Four studies evaluated the effects of SGLT-2 inhibitors on WC. Overall analysis showed that SGLT-2 inhibitors significantly reduced WC(WMD:-2. 20cm, 95%CI: -3. 81 to -0. 58, p = 0. 008). The $I^2$ was 0%, showing that the result was stable (Fig 2).

**VFA and SFA.** Eight studies of 429 participants showed that SGLT-2 inhibitors significantly decreased VFA compared with other antihyperglycemic drugs (MD:-14. 79cm$^2$, 95%CI: -24. 65 to -4. 93, p = 0. 003), with no heterogeneous($I^2$ = 0). Six studies evaluated the effects of SGLT-2 inhibitors on SFA. Overall analysis showed that SGLT-2 inhibitors significantly decreased SFA (WMD:-23. 27cm$^2$, 95% CI:-46. 44 to -0. 11, P = 0. 05), with no heterogeneity ($I^2$ = 0) (Fig 3).

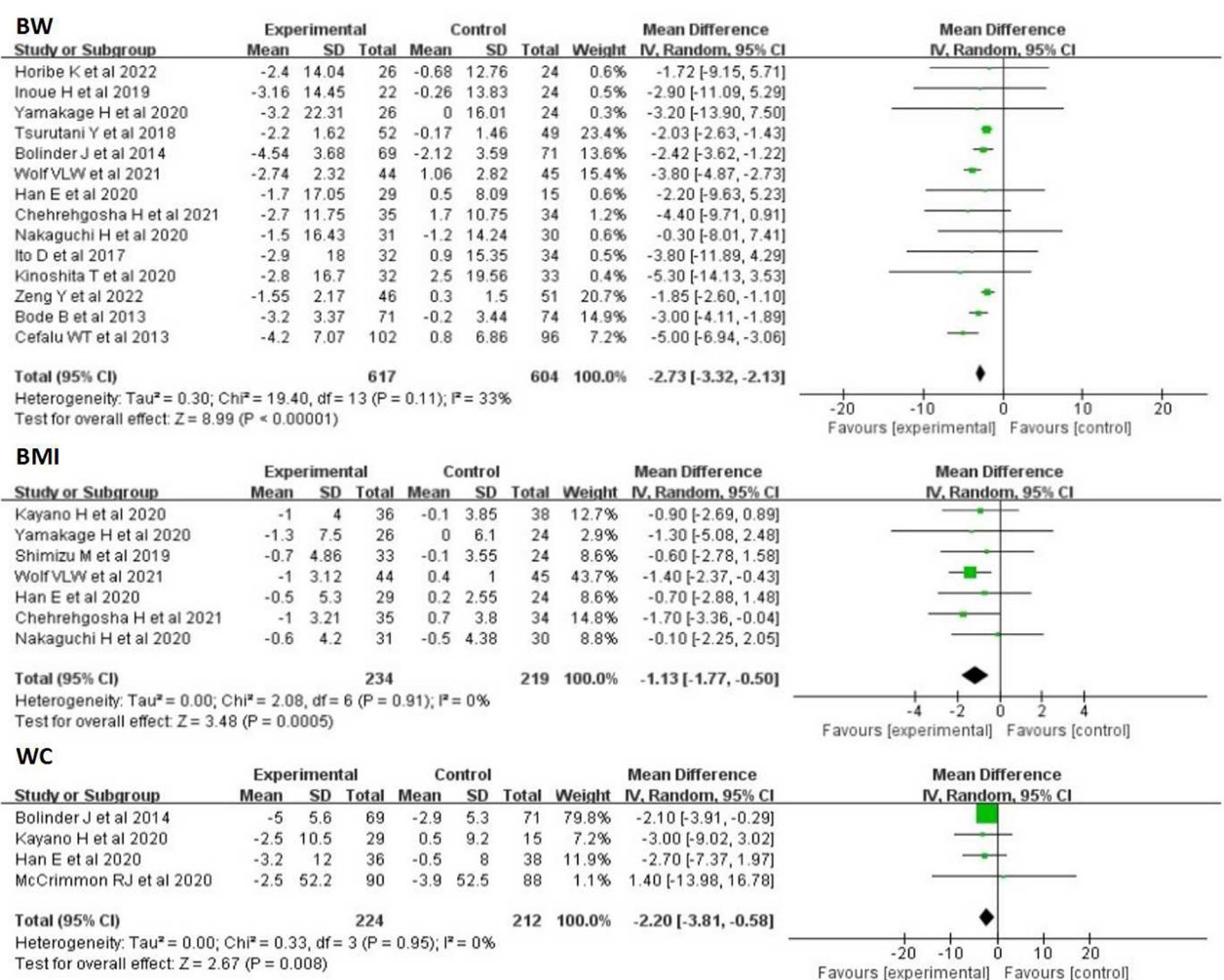

**Fig 2. Forest plots depicting BW, BMI and WC comparisons between SGLT-2 inhibitors and the control group.**

**FM, PBF, LM, SMM.** Ten studies of 827 participants showed that SGLT-2 inhibitors significantly decreased FM compared with other antihyperglycemic drugs (WMD:-1. 16kg, 95% CI: -2. 01 to -0. 31, p = 0. 008), with a moderate heterogeneous($I^2$ = 65%). There were four studies that reported the effect of SGLT-2 inhibitors on PBF. In contrast with the control group, SGLT-2 inhibitors evidently reduced PBF (WMD:-1. 50%, 95%CI:-2. 12 to -0. 87, P<0. 00001), with no heterogeneity($I^2$ = 0%). Nine studies evaluated the effects of SGLT-2 inhibitors on LM. Overall analysis showed that SGLT-2 inhibitors significantly decreased LM compared with other antihyperglycemic drugs (WMD:-0. 76kg, 95%CI:-1. 53 to 0. 01, P = 0. 05). A moderate heterogeneity was found between studies ($I^2$ = 74%). Seven studies in 206 SGLT-2 inhibitors users and 201 non-users evaluated the SMM. Overall analysis showed that SGLT-2 inhibitors significantly reduced SMM compared with other antihyperglycemic (WMD:-1. 01kg, 95%CI:-1. 91 to -0. 11, P = 0. 03), with no heterogeneity($I^2$ = 0) (Fig 4).

## Sensitivity analysis and subgroup analysis

In order to test heterogeneity, we did sensitivity analysis. When Nakaguchi and McCrimmon's studies were removed, the heterogeneity decreased from 65% to 29% in the analysis of FM,

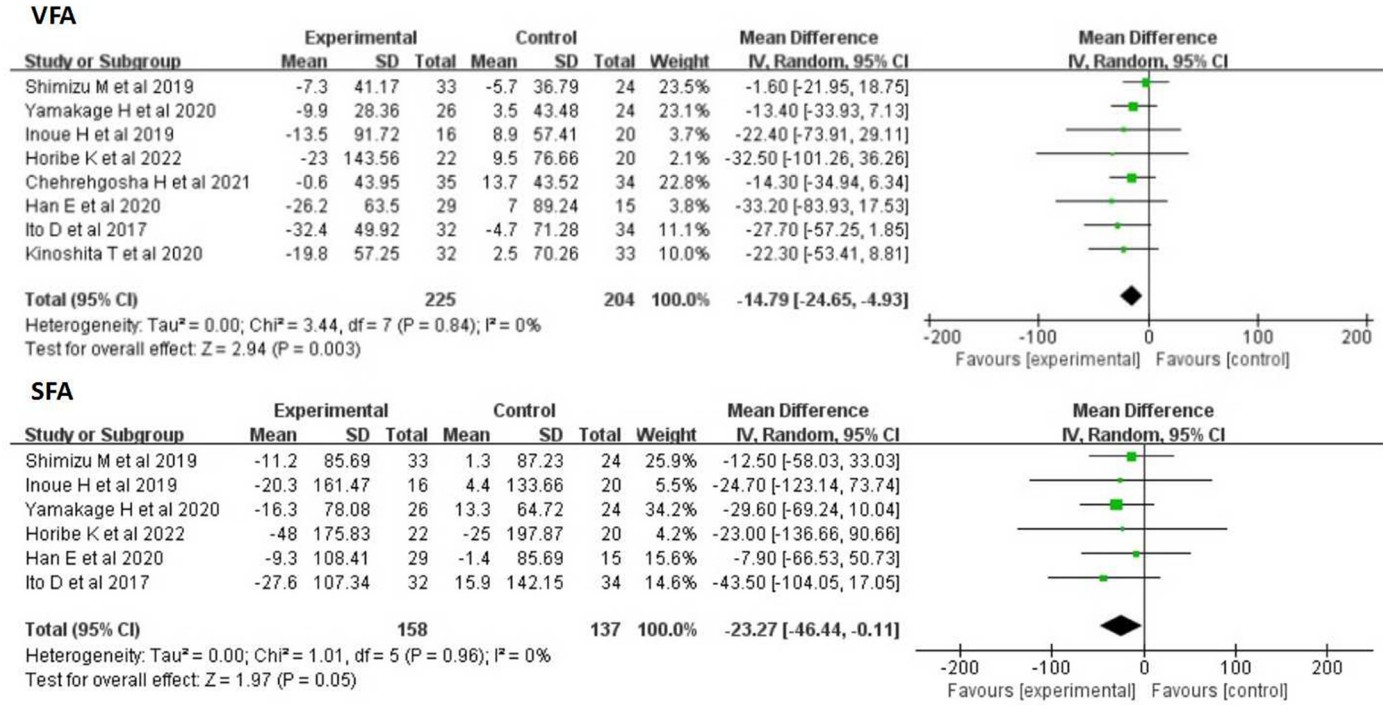

**Fig 3. Forest plots depicting VFA and SFA comparisons between SGLT-2 inhibitors and the control group.**

with a decrease in P values (from 0. 008 to less than 0. 0001), and the heterogeneity decreased from 74% to 51% in the analysis of LM, with a decrease in P values (from 0. 05 to 0. 0001), so we further performed a subgroup analysis based on different control groups. The results showed that SGLT-2 inhibitors significantly reduced FM compared with traditional hypoglycemic treatment (WMD:-1. 74kg, 95%CI:-2. 32 to -1. 16, p<0. 00001). SGLT-2 inhibitors seemed to reduce FM less than GLP-1RAs, but there was no statistical difference (WMD:0. 71kg, 95%CI:-0. 55 to 1. 97, P = 0. 27) (S3 Fig). Compared with traditional hypoglycemic agents, SGLT-2 inhibitors significantly reduced LM (WMD:-1. 13kg, 95%CI:-1. 71 to -0. 56, P = 0. 0001). Compared with GLP-1RAs, SGLT-2 inhibitors seemed to decrease LM to a lower extent, but there was no statistical difference (WMD:0. 76kg, 95%CI:-0. 04 to 1. 56, P = 0. 06) (S4 Fig). Other outcomes, such as BW, BMI, WC, VFA, SFA and SMM were not carried out further tests due to low heterogeneity.

## Discussion

This meta-analysis confirmed that SGLT-2 inhibitors reduced BW, BMI, WC, VFA, SFA, PBF, FM, LM and SMM to a greater extent than other hypoglycemic agents in T2DM.

Epidemiological evidence suggests that changes in body composition especially increased systemic FM and abdominal obesity, such as visceral adipose tissue, are strongly associated with T2DM risk [28]. BMI provides a fast and convenient standard for assessing obesity. However, the inability to specifically quantify total fat distribution, FM and LM has limited utility in estimating the risk of T2DM and other obesity-related diseases [29]. With similar BMI, LM decreased and FM increased more significantly in diabetic patients than in non-diabetic patients [30]. Studies have shown a strong relationship between T2DM and SO [4, 31], and body composition measurements are important tools for assessing both diseases [5, 6].

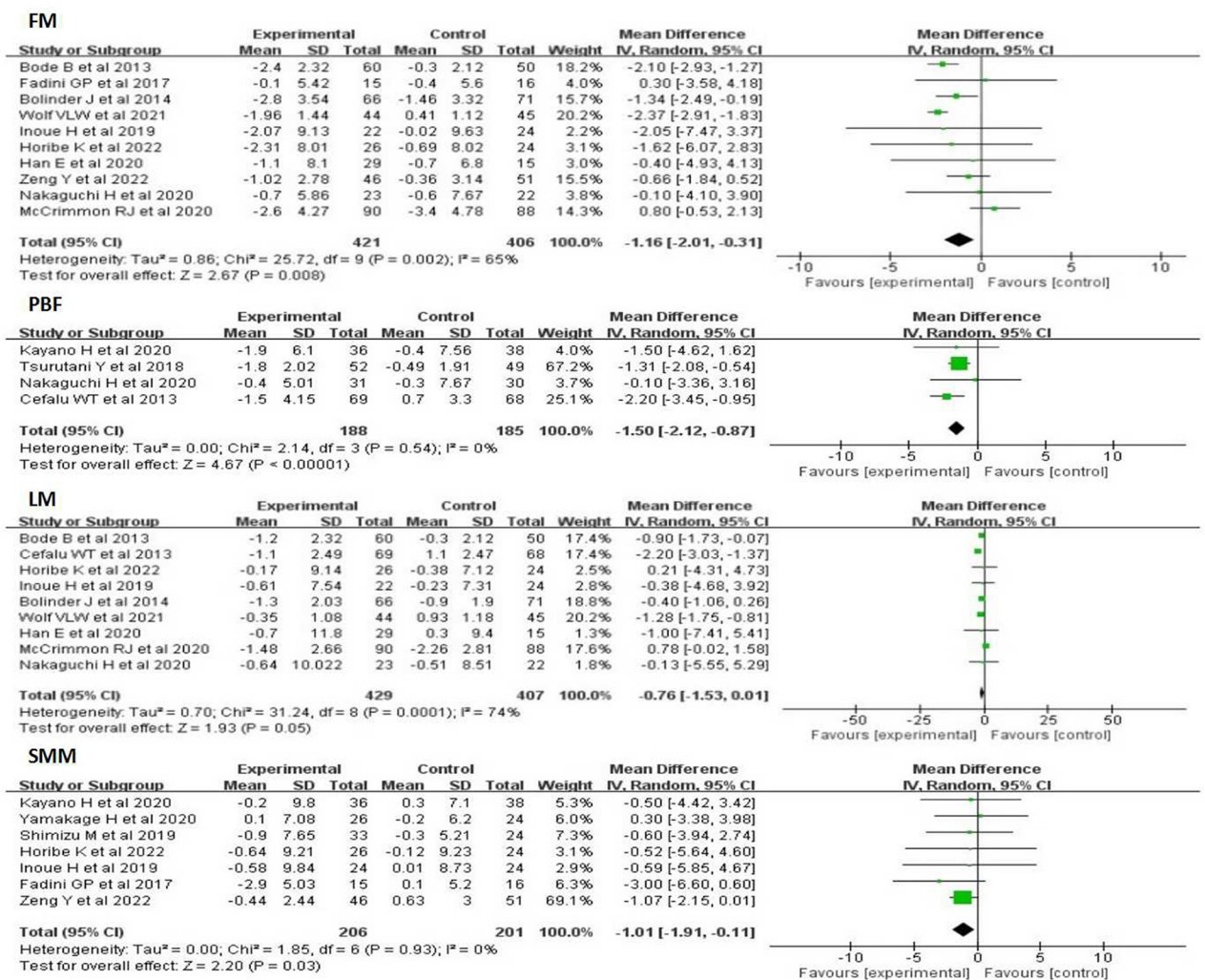

**Fig 4. Forest plots depicting FM, PBF, LM and SMM comparisons between SGLT-2 inhibitors and the control group.**

Bidirectional interactions have been hypothesized in obesity, low-grade inflammation, insulin resistance and sarcopenia [4]. Insulin in diabetic patients loses its function of promoting cellular glucose uptake and utilization, leading to insulin resistance, which destroys the role of insulin in inhibiting muscle protein breakdown, resulting in muscle fiber atrophy, muscle mass reduction and muscle strength decline [4]. On the contrary, SO may have synergistic effects with low-grade inflammation, which will increase production and secretion of various inflammatory factors and change insulin sensitivity by triggering different key steps of insulin signaling pathway, leading to insulin resistance and increasing risk of diabetes [32].

SGLT-2 inhibitors not only directly lead to weight loss through urinary glucose excretion mechanisms, but also improve adipocyte dysfunction in visceral adipose tissue, leading to leptin, visfatin, plasminogen activator inhibitor-1 decreased, adiponectin level increased, effectively promote lipolysis, reduce visceral fat [7, 33]. However, the unique mechanism of SGLT-2 inhibitors results in decreased insulin and elevated glucagon, limiting liver and muscle absorption of glucose and amino acids, promoting hepatic glucongenesis and glycogenolysis, and accelerating protein breakdown in muscle [9]. Case report of sarcopenia caused by SGLT-

2 inhibitor use in elderly patients with T2DM has been published [34]. Therefore, it is critical to evaluate the effect of SGLT-2 inhibitors on body composition, such as fat distribution, fat mass and muscle mass in patients with T2DM.

As expected, the role of SGLT-2 inhibitors in body weight, WC, VFA and SFA reduction was confirmed in this meta-analysis, which was consistent with the conclusions of previous two meta-analyses [35, 36]. However, the previous two studies only targeted T2DM patients with non-alcoholic fatty liver disease, while this study targeted a wider range of T2DM patients, and the number of articles and subjects included was larger. In addition, this meta-analysis confirmed the advantages of SGLT-2 inhibitors in reducing FM and BFP, which had not been mentioned in previous meta-analyses. To sum up, these results suggest that SGLT-2 inhibitors may play an important role in improving body composition. Cefalu et al. confirmed that about two-thirds of the weight loss caused by SGLT-2 inhibitors was attributable to a reduction in FM, with the remaining third attributable to a reduction in LM [11]. Due to limited data, the proportion of weight loss from LM was not analyzed in this meta-analysis. However, according to the number of FM, LM and SMM decrease(-1. 74kg, -1. 13kg, -1. 01kg) comparing with other traditional hypoglycemic drugs, it can be inferred that the reduction of FM accounted for a greater proportion in weight loss of SGLT-2 inhibitors, which was roughly consistent with Cefalu's conclusion. It should be noted that both LM and SMM play important roles in the diagnosis of sarcopenia and measurement of muscle mass [37]. LM measured with dual energy x-ray absorptiometry includes muscle, organs and body water, whereas bioelectrical impedance analysis measures SMM. This meta-analysis showed that SGLT-2 inhibitors significantly reduced both LM and SMM compared with other traditional hypoglycemic treatments, suggesting the adverse effects of SGLT-2 inhibitors on muscle mass decrease. In addition, the weight loss caused by SGLT-2 inhibitors may be due in part to body water loss, based on a unique glucose-lowering mechanism [38]. However, it should be noted that studies of ketosis and euglycemic ketoacidosis based on dehydration and insulinopenia during SGLT-2 inhibitors use have been reported [34, 39, 40]. Due to limited data, we did not conduct further analysis of water loss and risk of ketosis caused by SGLT-2 inhibitors. However, dehydration and ketosisis are also potential adverse effects that needs to be considered during SGLT-2 inhibitors use. In a word, despite producing a more favorable body composition, the potential of muscle mass loss induced by SGLT-2 inhibitors is noteworthy. Strategies to conserve skeletal muscle and improve physical function, such as through organized exercise, are important during the SGLT-2 inhibitors using.

Both GLP-1RAs and SGLT-2 inhibitors had favorable effects on BW. In this meta-analysis, the results showed that SGLT-2 inhibitors and GLP-1RAs had no significant difference in BW and FM loss. In addition, although there is no statistical difference in the LM reduction between the two drugs, the reduction by GLP-1RAs is more than that by SGLT-2 inhibitors. Therefore, the adverse effects of muscle mass reduction should be taken into account when the two drugs are used in patients with T2DM. However, this meta-analysis included only two studies comparing the body composition of the two drugs [16, 17]. More studies are needed to compare the difference in body composition changes between these two drugs.

The highlight of this meta-analysis is to confirm not only the advantages of SGLT-2 inhibitors in improving body composition, such as weight loss, BMI, WC, VFA, SFA, PBF and FM reduction, but also the adverse effects of these drugs on muscle mass reduction. However, the following are the limitations of this article:First, only a few RCTs met the conditions, and most of them had small sample sizes. Additional RCTs are needed to further validate the current results. Second, the included studies were followed up for a short period of time, with a median of 24 weeks, and the long-term effects of SGLT-2 inhibitors are unknown, so follow-up is needed. Third, due to the lack of data, the changes in the ratio of LM to FM and the changes in segmental LM were not analyzed. More research on the above parameters are needed to carry

out in-depth discussions. Fourth, the studies included in the meta-analysis are from different ethnic groups, age ranges, genders and patient groups with comorbidities, and that there are many parameters that may affect muscle loss, suggesting that pre-planned prospective control and large-scale studies will be more instructive in this regard.

## Conclusion

SGLT-2 inhibitors improve body composition in T2DM such as weight loss, BMI, WC, VFA, SFA, FM and PFM reduction, but cause adverse effects of reducing muscle mass. Therefore, until more evidence is obtained to support that SGLT-2 inhibitors increase the risk of sarcopenia, not only the benefit on body composition, but also the adverse effects of the reduction on muscle mass by SGLT-2 inhibitors in T2DM should be considered.

## Supporting information

**S1 Fig. Risk of bias graph.**
(TIF)

**S2 Fig. Risk of bias summary.**
(TIF)

**S3 Fig. Meta-analysis of FM level comparisons between SGLT-2 inhibitors and the control group based on the control.**
(TIF)

**S4 Fig. Meta-analysis of LM level comparisons between SGLT-2 inhibitors and the control group based on the control.**
(TIF)

**S1 Checklist. PRISMA 2009 checklist.**
(PDF)

## Author Contributions

**Conceptualization:** Runzhou Pan, Yongcai Zhao.

**Data curation:** Runzhou Pan.

**Formal analysis:** Runzhou Pan, Yan Zhang, Rongrong Wang.

**Investigation:** Yao Xu, Hong Ji.

**Methodology:** Runzhou Pan, Yongcai Zhao.

**Software:** Runzhou Pan, Yan Zhang.

**Supervision:** Yongcai Zhao.

**Visualization:** Yao Xu, Hong Ji.

**Writing – original draft:** Runzhou Pan.

**Writing – review & editing:** Runzhou Pan, Yongcai Zhao.

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
