## [Decision Letter · Decision Letter 0]

23 Nov 2022

PONE-D-22-30100Effect of SGLT-2 inhibitors on body composition in patients with type 2 diabetes mellitus: A meta-analysis of randomized controlled trialsPLOS ONE

Dear Dr. zhao,

Thank you for submitting your manuscript to PLOS ONE. After careful consideration, we feel that it has merit but does not fully meet PLOS ONE’s publication criteria as it currently stands. Therefore, we invite you to submit a revised version of the manuscript that addresses the points raised during the review process.

We look forward to receiving your revised manuscript.

Kind regards,

Gulali Aktas

Academic Editor

PLOS ONE

Additional Editor Comments:

The paper is somewhat interesting but it should be improved according to the suggestions of the reviewers before further process.

Reviewers' comments:

Reviewer's Responses to Questions

**Comments to the Author**

1. Is the manuscript technically sound, and do the data support the conclusions?

Reviewer #1: Yes

Reviewer #2: Yes

2. Has the statistical analysis been performed appropriately and rigorously? 

Reviewer #1: Yes

Reviewer #2: Yes

3. Have the authors made all data underlying the findings in their manuscript fully available?

Reviewer #1: Yes

Reviewer #2: Yes

4. Is the manuscript presented in an intelligible fashion and written in standard English?

Reviewer #1: Yes

Reviewer #2: Yes

5. Review Comments to the Author

Reviewer #1: Abstract is enough, The introduction is clear and explanatory. material method, statistics part is well explained and designed. The findings are beautifully and clearly written, both in writing and figures. In the literature, studies related to sglt2 have been scanned and discussed in a nice and detailed way. The discussion section could be expanded a bit though. You can benefit from the article mentioned in the journal below. I would also expect an important side effect such as euglycemic diabetic ketoacidosis to be mentioned in the discussion. In this respect, the article in the journal number 2 can be used.

1.Irish Journal of Medical Science (1971-), 191(4)

2.Journal of the College of Physicians and Surgeons--Pakistan: JCPSP, 32(7)

Reviewer #2: Dear author, the meta-analysis 'Effect of SGLT-2 inhibitors on body composition in patients with type 2 diabetes mellitus: A meta-analysis of randomized controlled trials' ;was evaluated. In this meta-analysis, the contribution of a group of drugs that are currently at the top of the preventive and therapeutic guidelines of all cardiovascular, diabetes and kidney diseases to muscle loss is evaluated.

I believe that it will contribute to the literature because it is a remarkable point.

Limitation: The fact that the studies included in the meta-analysis are from different ethnic groups, age ranges, genders and patient groups with comorbidities, and that there are many parameters that may affect muscle loss suggest that it is not appropriate to evaluate the results only for SGLT-2 treatment. Pre-planned prospective control and large-scale studies will be more instructive in this regard.

6. PLOS authors have the option to publish the peer review history of their article (what does this mean?). If published, this will include your full peer review and any attached files.

Reviewer #1: No

Reviewer #2: **Yes: **Pinar Yildiz

---

## [Author Response · Author response to Decision Letter 0]

2 Dec 2022

Responses to the comments of Reviewer #1

I have accepted all your suggestions and add the following in the discussion section:In addition,the weight loss caused by SGLT-2 inhibitors may be due in part to body water loss,based on a unique glucose-lowering mechanism[38].However,it should be noted that studies of ketosis and euglycemic ketoacidosis based on dehydration and insulinopenia during SGLT-2 inhibitors use have been reported[34,39,40].Due to limited data,we did not conduct further analysis of water loss and risk of ketosis caused by SGLT-2 inhibitors.However,dehydration and ketosisis are also potential adverse effects that needs to be considered during SGLT-2 inhibitors use.

References

[38] Post A, Groothof D, Eisenga MF, Bakker SJL. Sodium-Glucose Cotransporter 2 Inhibitors and Kidney Outcomes: True Renoprotection, Loss of Muscle Mass or Both? J Clin Med.2020. 9(5): 1603. DOI: 10.3390/jcm9051603. PMID: 32466262.

[39] Bilgin S, Duman TT, Kurtkulagi O, Yilmaz F, Aktas G. A Case of Euglycemic Diabetic Ketoacidosis due to Empagliflozin Use in a Patient with Type 1 Diabetes Mellitus. J Coll Physicians Surg Pak. 2022. 32(7): 928-930. DOI: 10.29271/jcpsp.2022.07.928. PMID: 35795946.

[40] Bilgin S, Kurtkulagi O, Duman TT, Tel BMA, Kahveci G, Kiran M, et al. Sodium glucose co-transporter-2 inhibitor, Empagliflozin, is associated with significant reduction in weight, body mass index, fasting glucose, and A1c levels in Type 2 diabetic patients with established coronary heart disease: the SUPER GATE study. Ir J Med Sci. 2022. 191(4): 1647-1652. DOI: 10.1007/s11845-021-02761-6. PMID: 34476725.

Responses to the comments of Reviewer #2

I have accepted all your suggestions and add the following in the discussion section:Fourth,the studies included in the meta-analysis are from different ethnic groups, age ranges, genders and patient groups with comorbidities, and that there are many parameters that may affect muscle loss,suggesting that pre-planned prospective control and large-scale studies will be more instructive in this regard.

---

## [Decision Letter · Decision Letter 1]

19 Dec 2022

Effect of SGLT-2 inhibitors on body composition in patients with type 2 diabetes mellitus: A meta-analysis of randomized controlled trials

PONE-D-22-30100R1

Dear Dr. zhao,

We’re pleased to inform you that your manuscript has been judged scientifically suitable for publication and will be formally accepted for publication once it meets all outstanding technical requirements.

Kind regards,

Gulali Aktas

Academic Editor

PLOS ONE

Additional Editor Comments (optional):

The manuacript is improved and reviewers suggest publication. I think the paper is acceptable, too. Well done.

Reviewers' comments:

Reviewer's Responses to Questions

**Comments to the Author**

1. If the authors have adequately addressed your comments raised in a previous round of review and you feel that this manuscript is now acceptable for publication, you may indicate that here to bypass the “Comments to the Author” section, enter your conflict of interest statement in the “Confidential to Editor” section, and submit your "Accept" recommendation.

Reviewer #1: All comments have been addressed

2. Is the manuscript technically sound, and do the data support the conclusions?

Reviewer #1: Yes

3. Has the statistical analysis been performed appropriately and rigorously? 

Reviewer #1: Yes

4. Have the authors made all data underlying the findings in their manuscript fully available?

Reviewer #1: Yes

5. Is the manuscript presented in an intelligible fashion and written in standard English?

Reviewer #1: Yes

6. Review Comments to the Author

Reviewer #1: Dear authors

I reviewed your article again. Thank you for reviewing the journals I recommended and mentioning the topics that are suitable for your study.

7. PLOS authors have the option to publish the peer review history of their article (what does this mean?). If published, this will include your full peer review and any attached files.

Reviewer #1: No

---

## [Editor Report · Acceptance letter]

21 Dec 2022

PONE-D-22-30100R1 

Effect of SGLT-2 inhibitors on body composition in patients with type 2 diabetes mellitus: A meta-analysis of randomized controlled trials 

Dear Dr. Zhao:

I'm pleased to inform you that your manuscript has been deemed suitable for publication in PLOS ONE. Congratulations! Your manuscript is now with our production department. 

Kind regards, 

on behalf of

Professor Gulali Aktas 

Academic Editor

PLOS ONE